# Malaria prevalence and use of control measures in an area with persistent transmission in Senegal

**Fassiatou Tairou**[1]*, **Ibrahima Gaye**[2], **Samantha Herrera**[3], **Saira Nawaz**[4], **Libasse Sarr**[5], **Birane Cissé**[5], **Babacar Faye**[1], **Roger C. K. Tine**[1]

**1** Department of Medical Parasitology, University Cheikh Anta Diop of Dakar, Dakar, Senegal, **2** Institut en Santé et Développement, University Cheikh Anta Diop of Dakar, Dakar, Senegal, **3** Malaria & Neglected Tropical Diseases Division, PATH, Washington, District of Columbia, United States of America, **4** Primary Health Care, PATH, Seattle, Washington, United States of America, **5** Department of Geography, University Cheikh Anta Diop of Dakar, Dakar, Senegal

* fassiatht@yahoo.fr

## Abstract

### Introduction

In Senegal, the widespread use of vector control measures has resulted in a significant reduction in the malaria burden and led the country to consider the possibility of elimination. Given this shift and changing context, it is important to characterize the malaria burden across all age groups to guide decision-making on programmatic interventions to interrupt transmission and ultimately eradicate the disease. In Senegal, there is a lack of information on malaria prevalence among certain populations, particularly among adolescents and adults. This study sought to assess the magnitude of malaria infections in all age groups, as well as malaria associated factors in an area of persistent transmission in Senegal.

### Methods

A cross-sectional household survey was conducted in four health posts (Khossanto, Mamakhona, Diakhaling and Sambrambougou), of the health district of Saraya, in November 2021, among individuals over 6 months of age. Households were selected using multistage sampling. Consented participants were screened for malaria parasites by microscopic examination of blood smears, and hemoglobin levels were measured using the Hemocue HB 301™ analyzer. Socio-demographic information of the participants, household heads, household assets, and information on ownership and use of preventive measures were collected using a structured questionnaire. Weighted generalized mixed effects logistic regression model was used to identify factors associated with microscopically confirmed malaria infection.

### Results

A total of 1759 participants were enrolled in the study. Overall, about 21% of participants were classified as having *Plasmodium* infection; children aged 5–10 years old (26.6%), adolescents aged 10–19 years old (24.7%), and children under five years of age (20.5%) had

**Data Availability Statement:** All relevant data are within the manuscript and its Supporting Information files

**Funding:** The study was supported by the Senegalese National Malaria Control Program, through an agreement with the Department of Parasitology and Mycology of University Cheikh Anta Diop of Dakar (# funding number: NFM 2-SEN-M-PNLP 2018 - 2019 -2020). The funder had no role in the study design, data collection, and analysis, decision to publish, or preparation of the manuscript.

**Competing interests:** The authors have declared that no competing interests exist

higher rates of infection compared to adults (13.5%). *Plasmodium falciparum* accounted for 99.2% of the malaria infections, and most infections (69%) were asymptomatic. Around one-third of study participants had anemia (hemoglobin level <11.0 g/dl), with under five children bearing the highest burden (67.3%). Multivariate analysis showed that the odds of having a malaria infection were around 2 times higher among participants in Khossanto compared to Diakhaling (aOR = 1.84, 95% CI:1.06–3.20). Participants aged 5–9 years were more likely to have malaria infection compared to under five children (aOR = 1.40, 95% CI:1.02–1.91). Factors associated with anemia were *P. falciparum* infection (aOR = 1.36, p = 0.027), females (aOR = 2.16, p = 0.000), under-five age group (aOR = 13.01, p = 0.000).

## Conclusion

Malaria burden was considerable among adolescents and under ten children living in an area of persistent transmission, with adolescents more commonly presenting as asymptomatic. Interventions tailored to this specific group of the population are needed to better control the disease and reduce its burden.

## Introduction

Malaria remains a major public health concern despite important progress made in reducing the disease burden over the past two decades. Sub-Saharan African countries bear the highest share of malaria, with about 94% (233 million) of all cases and 96% (580 000) of all malaria deaths in 2022 [1]. The COVID-19 pandemic has disrupted malaria-related services in many countries and led to an increase in the malaria burden globally, though this has been more predominant in sub-Saharan Africa [2]. Between 2019 and 2020, malaria incidence increased from 222 to 232 per 1000 population at risk in the World Health organization (WHO) African Region and the mortality rate from 57 to 61 deaths per 100 000 population at risk [2]. In 2022, malaria incidence declined to 223 per 1000 population at risk, and the mortality rate to 56 deaths per 100 000 population at risk [1].

In Senegal, malaria is endemic in most of the country, with an upsurge during the rainy season [3]. The disease is unevenly distributed between the fourteen regions of the country. Regions located in the southern part (Kolda, Tambacounda, and Kedougou) recorded 83.3% of cases and 51% of malaria-related deaths in 2020 [4]. Several malaria interventions are being implemented in Senegal including i) diagnosis and treatment using rapid diagnostic tests (RDTs)/microscopy and Artemisinin-based Combination Therapies (ACTs), ii) community case management ("Prise en charge à domicile"- PECADOM and PECADOM plus in specific districts), (iii) vector control with long-lasting insecticidal nets (LLINs) and indoor residual spraying (IRS) in targeted districts, iv) seasonal malaria chemoprevention (SMC) for children under 10 years of age living in eligible areas according to WHO criteria, and (iv) intermittent preventive treatment (IPT) with Sulfadoxine Pyrimethamine (SP) for pregnant women [5,6].

The progress made in the fight against malaria has contributed to a nationwide decrease of malaria burden in Senegal, despite a slight increase in 2020 due to the Covid-19 pandemic [3,6]. Between 2015 to 2019, proportional morbidity (proportion of all-cause consultations associated with parasitologically confirmed malaria) and mortality (proportion of all hospitalized deaths associated with parasitologically confirmed malaria) [5] have decreased from 4.86% and 3.52% to 3.03% and 1.66%, respectively [7]. Due to the substantial reduction in

malaria burden, Senegal has outlined a vision of malaria elimination by 2030 in parts of the country where malaria morbidity has fallen sharply and remained at a low level [3]. Senegal captures information on malaria prevalence in children under five and pregnant women through national surveys, including the continuous Demographic and Health Survey (cDHS) and the Malaria Indicator Survey (MIS) [8,9]; however, there is very limited malaria epidemiological data available for all age groups. In addition, these surveys are designed to provide malaria prevalence estimates at a regional level. To guide elimination strategies and intervention planning at subnational level, more granular data are needed.

As Senegal shifts from a control phase to a context of malaria elimination, a better and more granular understanding of the disease pattern across the population is needed, to help support a more targeted approach. In this study, a district-level cross-sectional survey was conducted to assess the magnitude of malaria infection in all age groups, as well as malaria-associated factors in an area of persistent transmission in Senegal.

## Methods

### Study design and population

A cross-sectional, community-based survey was conducted from 03rd to 18th November 2021, in the health district of Saraya, among individuals over 6 months of age. Individuals who reported living in the study area for at least six months, and who have provided written informed consent (or parents/caregivers consented for the participation of children) were eligible to participate. People who reported a malaria episode and treated with artemisinin-based combination therapy (ACT) in three weeks preceding the study were not enrolled.

### Study site

The health district of Saraya is located in the Kedougou region in the south-eastern part of Senegal, at 800 km from the capital, Dakar. The district shares a border with Mali on the east, Guinea in the south, the region of Tambacounda to the north, and the health district of Kedougou to the west. The health district includes 102 villages and occupies a land area of 7803 km$^2$, with an estimated population of 66,017 inhabitants in 2021 [10]. The district has 1 health center, 22 health posts, 28 health huts, and 78 villages with a village volunteer for malaria case management (named "DSDOM", *Dispensateur de soins à domicile*). The climate is Sudano-Guinean with a dry season and a rainy season. Malaria is meso endemic and stable in Saraya, with a long transmission season from July to December. Transmission intensity remains high during the rainy season, with a high biting rate (25 bites/person/night) [11] and high morbidity during the transmission period. The major vectors are *Anopheles gambiae*, *An. arabiensis sl*, *An. funestus and An. nili* [12]. Malaria incidence (per 1000 population at risk) was 379.8 in 2019, 607.6 in 2020, and 744.7 in 2021 [4,6]. Within Saraya health district, the study was carried out in the catchment areas of the four health posts that reported the highest number of malaria cases during the previous transmission season in 2020 (*Diakhaling*, *khossanto*, *Mamakhono*, *Sambrambougou health posts*) (Fig 1).

### Sample size and sampling method

Assuming a malaria prevalence of 7% among children under five years of age in the study area [9], a sample of 1000 children was calculated to provide a statistical power of 90% and to detect a difference of 3% with an alpha risk of 5% in a two-sided situation.

Thirty clusters represented by fifteen villages in the targeted health posts were selected using a two-level random cluster sampling approach with probability proportional to size in

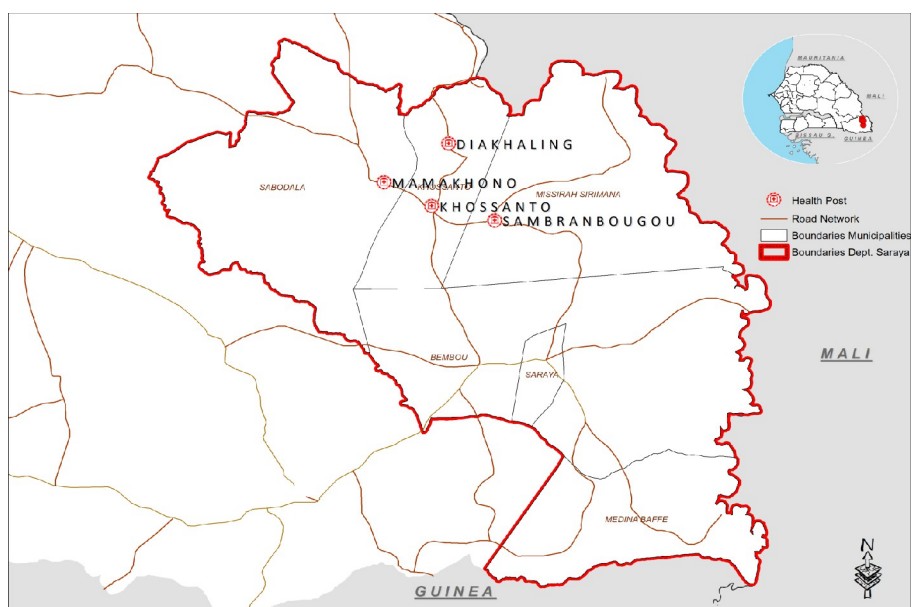

**Fig 1. Health district of Saraya showing the study health posts.** Reprinted from [SARR and al. 2023] under a CC BY license, with permission from [LIBASSE HANE SARR M.], original copyright [2023].

the villages. After the selection of the clusters, households were selected using the compact segment sampling technique. A map of each selected village was drawn and divided into segments, so that each segment included approximately 63 persons aged 6 months or older (in total, in all age groups) including 33 persons aged under 20 years old. The number of segments in each village was calculated as the approximate village total population divided by 63, rounded down to the nearest whole number [13,14]. The segment to be surveyed was then randomly selected, its boundaries were transformed in shape file and integrated in the LOCUS MAP navigation software to be easily identified in the field. All the households in the segment were visited and all the individuals over 6 months of age who provided written informed consent for study participation were enrolled in the survey. Written informed consent was obtained from parents for participants below the age of 18 years old.

The number of persons to be recruited from each of the following age groups (6 months-9 years, 10 to 19 years, and >20 years) was determined by applying the percentage of representability of each age group in the Senegalese population [15] to the sample size calculated.

## Data collection

The study questionnaire was developed in French and deployed in an electronic form using Open Data Kit (ODK), and included information on the demographics (age, gender, village, ethnicity, education level, and occupation) of the participants and those of their household head (age, gender, occupation, education level), the household characteristics (GPS coordinates, type of wall, roof, floor, water source, type of toilet), as well as the assets of the household. In addition, data on the travel history of the participants outside the village in the previous four weeks and participants' use of malaria preventive methods (bed net ownership and usage, use of other malaria preventive measures such as coils, repellents, wearing of long clothes, hand cream, cleaning of sewage) was collected. Furthermore, information on the receipt of seasonal malaria chemoprevention (SMC) during the 2021 campaign (four rounds) was collected for children aged 6 months to 10 years, based on caregiver/mother report. The

questionnaire was piloted in Saraya city in an area that was not included in the survey, to check for understanding and accuracy of the questions by the population and was then adapted in accordance with findings from the piloting. The questionnaire was administered to the participants by trained enumerators, using android tablets. The main outcome measures included the prevalence of malaria parasitemia and anemia.

## Sample collection and laboratory methods

The team in charge of the sample collection and the preparation of the blood smears was composed of two experienced laboratory technicians from the department of medical parasitology of Université Cheikh Anta Diop (UCAD) of Dakar, supported by trained community health worker. A blood sample was taken from each participant by finger prick. Two slides with both thick and thin smears were prepared for each participant. The slides were dried at room temperature, and the thin smears were fixed with methanol. Both thin and thick smears were stained with 10% Giemsa for 15 minutes on the site to ensure their quality and integrity. The slides were then read at the laboratory of medical parasitology of UCAD (the National Malaria Reference Laboratory) by experienced microscopists. Each slide was read by two independent microscopists. The species (*Plasmodium (P). falciparum*, *P. malariae*, *P. vivax*, *or P. ovale)* and the stade *(*asexual and sexual) for each positive slide were determined. The parasite density (per μl) was assessed for asexual parasites by counting the number of parasites per 200/500 white blood cells, as follows: (Number of parasites counted X 8000)/number of leukocytes counted. In case of a discrepancy between the two readings (positive versus negative or a difference of more than 10% on the parasite density), a 3rd reading was performed by another microscopist. The final parasite density was the average of the parasite densities of the two closer readings.

A malaria rapid diagnosis test (RDT) was performed for those with fever (axillary temperature ≥37.5˚C) or a history of fever in the previous 48 hours, using *P. falciparum* histidine-rich protein 2 (PfHRP2) test (mRDT- SD Bioline).

Hemoglobin level was measured for each participant using a portable hemoglobinometer. A drop of finger prick blood was drawn into a microcuvette for hemoglobin determination using the HemoCue® machine (HemoCue® AB Hb 301, Ängelholm, Sweden).

## Data management and statistical analysis

A unique identifier was attributed to each participant and used to label the samples. The data collected were checked for completeness and consistency. Cleaned data were analyzed using STATA software (Stata Corp 2016). Continuous variables were summarized into means ± standard deviation (SD), and categorical variables into frequencies and proportions. Means were compared using Student's t-test and proportions using Pearson's Chi-square test.

A multistage sampling technique, known as a complex survey design was used in this survey. Thus ordinary logistic regression is not appropriate for the modeling of such survey data [16–18]. To take into consideration the complexity of the survey design, a weighted generalized mixed effects logistic regression model was used to explore potential determinants of malaria and factors associated with anemia among the study population. All two-way interactions formed by the independent variables were assessed. Significant variables with p≤0.25 in univariate analysis were introduced in the multivariate analysis. Calibration statistics were computed to assess the model fit, and the R-squared was 0.94, indicating a good model fit [19,20]. The selection of the final model was based on criteria such as Akaike's information criterion (AIC) and Bayesian information criterion (BIC) [16,18]. The significance level for the final model was set at 5% (two-sided).

Response variable: the dependent variable was binary, it referred to the result of microscopy and indicated whether the participant was tested positive (1) or negative (0) for malaria.

Independent variables: explanatory variables consisted of the socio-demographic characteristics of the participant (age, gender), those of the household head (education level, wealth index, gender), household characteristics and assets (e.g., type of wall, type of floor, type of roof, household ownership of different materials or goods), the number of persons in the household, possession of bed net and its use the night before the survey, the location of the health post, history of travel outside of the village.

The primary outcome was the prevalence of *P. falciparum* infection confirmed by microscopy. Secondary outcomes included: the prevalence of symptomatic malaria, the prevalence of asymptomatic malaria, and the prevalence of anemia.

**Definition of terms.** Asymptomatic malaria was defined as individuals who were microscopy positive, with a temperature below 37.5°C and absence of malaria-related symptoms [21].

Symptomatic malaria was defined as individuals with microscopically confirmed infection and an axillary temperature >37.5°C [22].

Anemia was defined as a hemoglobin count less than 11 grams per deciliter (g/dl) and was categorized as severe (Hb level < 8.0 g/dl), and moderate (between 8 and 11 g/dl) [23].

Parasite density was categorized as low (< 5000) and high (≥ 5000) density.

Full SMC coverage was defined as proportion of children aged 6 months to 10 years who received the complete 3 days treatment course for all four rounds.

The wealth index was calculated based on the household assets (water source, type of toilet, ownership of radio, TV, bicycle, fridge, combustible) and household characteristics (type of wall, type of floor, type of roof), using a principal component analysis [24,25]. The index was then categorized into five levels (highest, fourth, middle, second, lowest).

## Ethical considerations

The study protocol was approved by the Institutional Ethics Committee of University Cheikh Anta Diop of Dakar (CER/UCAD/AD/MSN /039/2020). Administrative authorization was sought from the health district of Saraya, and from the community's leaders to conduct the study in their area and for the participation of the community. Participation in the study was voluntary. Prior to each participant's enrollment, written informed consent was sought from the parents/guardians of all children, aged 6 months to 17 years old for their child's participation. In addition, written assent was sought from participants 15–17 years old of age. Furthermore, older adolescents and adults, aged 18 years old and more were invited to provide written informed consent to participate. No personal identifiable data was collected; a unique identifier was attributed to each participant. All the data collected for the study was kept confidential, and used only for the study purpose, and access was restricted to the study staff. During the course of the survey, participants who were found with moderate anemia (hemoglobin <11 g/dl) were treated as per national guidelines, and those with severe anemia (hemoglobin <8g/dl) were referred to the district health center for treatment. Individuals with positive RDTs were treated, with artemether-lumefantrine (AL) according to the national guidelines. The required treatments for patients with malaria and anemia were covered by the study.

## Results

### Characteristics of the study participants

A total of 1759 participants were enrolled in the study, among which 22.1% were from the Diakhaling health post, 30.8% from Khossanto, 27.8% from Mamakhono and 19.3% from Sambrambougou. Twenty percent of the participants were below 5 years of age, 23.7% were 5–9

years old, 13.2% were 10–14 years old, 9.1% were 15–19 years old, and 33.7% were 20 years old and above. Just over half of the participants were female (54.1%); the Malinke ethnicity represented 77.4% of the survey sample. Just under half of the participants had received no formal education (46.6%), or had no formal occupation (43%)

According to the report of caregivers/mothers, a proportion of sixty-eight percent (68.8%) of children aged 6 months to 10 years (n = 839) received at least one dose of SMC; full coverage of SMC (3 days of treatment for all four rounds) was evaluated at 46.8% (Table 1).

## Ownership, usage of LLINs, and other preventive measures

Overall, about half of participants (49.4%) reported that their household owned a LLIN; with household ownership of bed nets slightly lower among adolescent participants 15–19 years of age (40.6%) (Fig 2).

The percentage of participants who reported using a LLIN the night before the survey was 43.1%. LLIN use was reported to be the highest among participants below five years of age and 20 years of age and above (about 48%) (Fig 2).

Forty-two percent of the participants, reported using other malaria preventive measures. Of those that reported using other preventive measures, the most common measures included coils (66.4%), insecticide spray (32.1%), and wearing long clothes (26.9%).

## Prevalence of malaria parasitemia among the study participants

Overall, the prevalence of *Plasmodium* infection. was 20.5% among the study participants. The highest prevalence was observed among children 5–9 years of age (26.6%), followed by adolescents aged 10–19 years (24.7%) and under-five children (20.5%). Adults 20 years of age and above had the lowest parasite prevalence, at 13.5%.

The highest geometric mean density of *P. falciparum* parasites, 3322.4 (2202.0–5012.8) was observed among children aged 5–9 years old.

*P. falciparum* was the predominant parasite species, accounting for 99.2% of malaria infections. In addition to *P. falciparum*, three individuals among the study participants harbored other *Plasmodium* species, 2 had mixed infections (0.11%) with *P falciparum* and *P. malaria*, and 1 had a mixed infection (0.06%) with *P. falciparum* and *P. ovale*.

A total number of 43 individuals were found with gametocytes, providing a prevalence of gametocyte carriage at 2.4% among the study participants. Age distribution of gametocyte carriage was as follow: 2.8% for under five children, 3.4% for 5–9 years old children, 2.5% for 10–19 years old adolescents, and 1.5% among adults above the age of 20 years.

The overall prevalence of asymptomatic *Plasmodium* infection was 14.2% among the study participants. Asymptomatic *P*. infection was highest among children 5–9 years of age (17.0%) and adolescents 10–19 years of age (18.4%), compared to children under-five (11.8%) and adults 20 years of age and above (10.9%).

The prevalence of symptomatic *Plasmodium* infection was 6.3% in the study. Symptomatic *P*. infection was highest among children under-five of age (8.7%) and children 5–9 years of age (9.6%) (Table 2).

The prevalence of *P. falciparum* malaria infection was highest in Khossanto at 26.4% (22.5% -30.3%) and lowest in Diakhaling at 15.2% (11.9% -19.1%) (Fig 3).

## Prevalence of anemia among the study participants

Hemoglobin was measured for 1723 participants; out of these, 33.5% were anemic (Hb < 11g/dL), with 29.8% with moderate anemia and 3.8% with severe anemia. Under-five children had the highest rate of anemia (67.3%), compared to other age groups (Fig 4).

**Table 1. Socio-demographics characteristics of the study participants.**

| Variables | Overall, N = 1759 | 6 months to <5 years, n = 356 | 5 to <10 years, n = 417 | 10 to <15 years, n = 232 | 15 to 19 years, n = 160 | > = 20 years, n = 594 |
|---|---|---|---|---|---|---|
| **Characteristics of the participants** | | | | | | |
| **Sex** | | | | | | |
| Female | 951(54.1) | 177(49.7) | 219(52.5) | 119(51.3) | 101(63.1) | 335(56.4) |
| **Ethnicity** | | | | | | |
| Malinke | 1362(77.4) | 283(79.5) | 348(83.5) | 189(81.5) | 117(73.1) | 425(71.6) |
| Pular | 233(13.3) | 47(13.2) | 50(12.0) | 33(14.2) | 29(18.1) | 74(12.5) |
| Other* | 164(9.3) | 26(7.3) | 19(4.6) | 10(4.3) | 14(8.8) | 95(16.0) |
| **Education level** | | | | | | |
| None | 820(46.6) | 356(100.0) | 135(32.4) | 27(11.6) | 41(25.6) | 261(43.9) |
| Koranic school | 193(11.0) | -- | 34(8.2) | 29(12.5) | 23(14.4) | 107(18.0) |
| Primary | 650(36.9) | -- | 237(56.8% | 172(74.1) | 66(41.3) | 175(29.5) |
| Secondary | 82(4.7) | -- | -- | 4(1.7) | 30(18.8) | 48(8.1) |
| University | 03(0.2) | -- | -- | -- | -- | 03(0.5) |
| Other$^{\Psi}$ | 11(0.7) | -- | 11(2.6) | -- | -- | -- |
| **Occupation** | | | | | | |
| None | 759(43.2) | 356(100.0) | 140(33.6) | 47(20.3) | 67(41.9) | 149(25.1) |
| Farmer | 143(8.1) | -- | 03(0.7) | 02(0.9) | 12(7.5) | 126(21.2) |
| Student | 452(25.7) | -- | 261(62.6) | 160(68.9) | 28(17.5) | 03(0.5) |
| Gold digger | 276(15.7) | -- | 03(0.7) | 09(3.9) | 36(22.5) | 228(38.4) |
| Vendor/trader | 54(3.1) | -- | 01(0.2) | 06(2.6) | 04(7.4) | 40(6.8) |
| Worker | 52(2.9) | -- | 07(1.7) | 06(2.6) | 04(7.4) | 32(5.4) |
| Other# | 23(1.3) | -- | 02(0.5) | 02(0.9) | 03(1.9) | 16(2.7) |
| **History of travel in the previous 4 weeks** | 169(9.6) | 21(5.9) | 29(6.9) | 19(8.2) | 16(10.0) | 84(14.1) |
| **SMC receipt in 2021** | | | | | | |
| **Yes** | 577(68.8)$^{\Phi}$ | 244(68.5) | 304(72.9) | 29(43.9)$^{\delta}$ | -- | -- |
| **No** | 262(31.2)$^{\Phi}$ | 112(31.5) | 113(27.1) | 37(56.0)$^{\delta}$ | -- | -- |
| **SMC full treatment course** | 393(46.8)$^{\Phi}$ | 160(44.9) | 214(51.3) | 19(28.8)$^{\delta}$ | -- | -- |
| **Characteristics of the household's head** | | | | | | |
| **Age-means (SD)** | 45.5(15.5) | 45.6(15.6) | 47.8(14.0) | 49.4(13.8) | 44.2(16.0) | 43.0(16.3) |
| **Sex** | | | | | | |
| Male | 1641(93.3) | 325(91.3) | 393(94.2) | 217(93.5) | 147(91.9) | 559(94.1) |
| Female | 106(6.0) | 24(6.7) | 22(5.3) | 13(5.6) | 12(7.5) | 35(5.9) |
| Missing | 12(0.7) | 07(2.0) | 02(0.5) | 02(0.9) | 01(0.6) | -- |
| **Education level** | | | | | | |
| None | 700(39.8) | 141(39.6) | 161(38.6) | 100(43.1) | 64(40.0) | 235(39.6) |
| Koranic school | 588(33.4) | 119(33.4) | 138(33.1) | 79(34.1) | 56(35.0) | 196(33.0) |
| Primary | 426(24.2) | 83(23.3) | 113(27.1) | 48(20.7) | 33(20.6) | 149(25.1) |
| Secondary | 32(1.8) | 06(1.7) | 03(0.7) | 03(1.3) | 06(3.7) | 14(2.4) |
| University | 01(0.1) | -- | -- | -- | -- | 01(0.2) |
| Missing | 12(07) | 07(2.0) | 02(0.5) | 02(0.9) | 01(0.6) | -- |
| **Occupation** | | | | | | |
| None | 109(6.2) | 22(6.2) | 33(7.9) | 09(3.9) | 08(5.0) | 37(6.2) |
| Farmer | 762(43.3) | 160(45.0) | 200(48.0) | 104(44.8) | 65(40.6) | 233(39.2) |
| Gold digger | 617(35.1) | 118(33.2) | 123(29.5) | 59(36.9) | 136(34.7) | 240(40.4) |
| Vendor/trader | 80(4.6) | 15(4.2) | 18(4.3) | 13(5.6) | 08(5.0) | 26(4.4% |
| Worker | 100(5.7) | 20(5.6) | 22(5.3) | 13(5.6) | 09(5.6) | 36(6.1) |

*(Continued)*

**Table 1.** (Continued)

| Variables | Overall, N = 1759 | 6 months to <5 years, n = 356 | 5 to <10 years, n = 417 | 10 to <15 years, n = 232 | 15 to 19 years, n = 160 | > = 20 years, n = 594 |
|---|---|---|---|---|---|---|
| Other[$] | 79(4.5) | 14(3.9) | 19(4.6) | 14(6.0) | 10(6.3) | 22(3.7) |
| Missing | 12(0.7) | 07(2.0) | 02(0.5) | 02(0.9) | 01(0.6) | -- |
| **Wealth index** | | | | | | |
| Highest | 351(19.9) | 74(20.8) | 93(22.3) | 44(19.0) | 34(21.3) | 106(17.9) |
| Fourth | 352(20.0) | 77(21.6) | 90(21.6) | 47(20.3) | 29(18.0) | 109(18.4) |
| Middle | 352(20.0) | 68(19.1) | 99(23.7) | 52(22.4) | 31(19.4) | 102(17.2) |
| Second | 352(20.0) | 70(19.7) | 83(19.9) | 51(22.0) | 30(18.8) | 118(19.9) |
| Lowest | 352(20.0) | 67(18.8) | 52(12.5) | 38(16.4) | 36(22.5) | 159(26.8) |

*Dialounke, Bambara, Dogon, Bosso, Diola, Serere, Bassari, Wolof and Mossi.

Ψ Preschool.

# Household wife, driver, community health worker, Koranic teacher.

$ Electrician, community health worker, butcher, imam, tailor, village chief.

Φ Percentage based on the total number (839) of children aged ≤ 10 years.

δ Percentage based on the number (66) of children aged 10 years.

The prevalence of anemia was similar across the 4 health posts (ranging from around 28.8% in Sambrambougou to 35.1% in Mamakhono).

## Factors associated with malaria infection

In a multivariate analysis, children aged 5–9 years were 1.40 times more likely to have malaria compared to their under five years counterparts (aOR = 1.40, 95% CI: 1.02–1.91) while adults aged 20 years and over (≥20) were less likely to have malaria infection than children under five (aOR = 0.64, 95% CI: 0.46–0.90).

The odds of having malaria parasites was higher in the health post of Khossanto (aOR = 1.84, 95% CI: 1.06–3.20) compared to Diakhaling (Table 3).

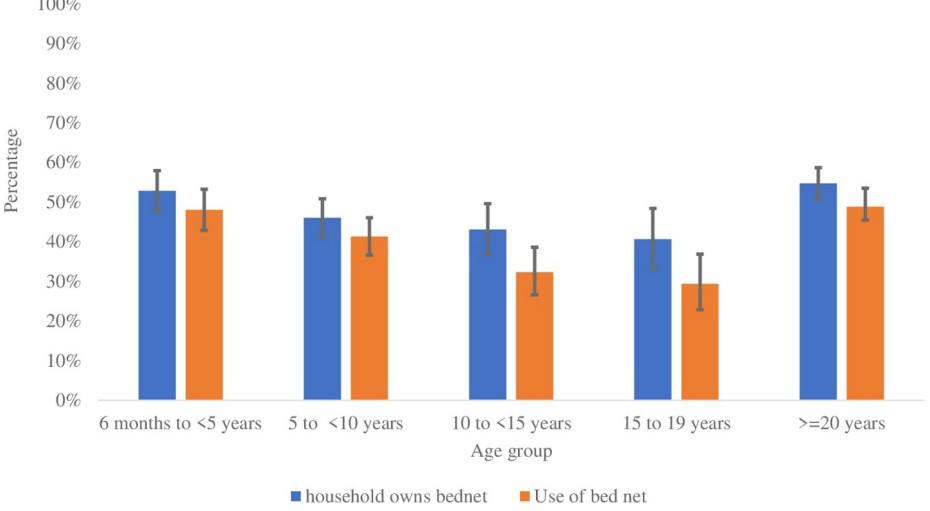

**Fig 2. Household bed net ownership and individual bed net use.**

**Table 2. Prevalence of malaria infection among the study participants by age group.**

| Parameters N (%) | Overall, N = 1759 | 6 months to <5 years, n = 356 | 5 to <10 years, n = 417 | 10 to <15 years, n = 232 | 15 to 19 years, n = 160 | > = 20 years, n = 594 |
|---|---|---|---|---|---|---|
| Fever (Temperature ≥ 37.5) | 289(16.9) | 77(21.60 | 75(17.9) | 37(15.9) | 19(11.9) | 81(13.6) |
| *Plasmodium* infection | | | | | | |
| *Plasmodium* (*P.*) infection | 361(20.5) | 73(20.5) | 111(26.6) | 57(24.6) | 40(25.0) | 80(13.5) |
| Asymptomatic *P.* infection | 250(14.2) | 42(11.8) | 71(17.0) | 37(15.9) | 35(21.9) | 65(10.9) |
| Symptomatic *P.* infection | 111(6.3) | 31(8.7) | 40(9.6) | 20(8.6) | 5(3.1) | 15(2.5) |
| *P. falciparum* | 358(20.4) | 72(20.2) | 111(26.6) | 56(24.1) | 39(24.4) | 80(13.5) |
| Mixed infection (*P falciparum*+ *P. malariae*) | -- | 01(0.3) | -- | 01(0.4) | -- | -- |
| Mixed infection (*P falciparum* + *P. Ovale*) | -- | -- | -- | -- | 01(0.6) | -- |
| Geometric means of *P. falciparum* density | 1440.9 (1146.9–1810.2) | 1553.64(918.2–2628.9) | 3322.36(2201.9–5012.8) | 1498.94(857.4–2620.7) | 542.66(300.1–981.2) | 643.65(416.7–994.2) |
| Prevalence of *P. falciparum* gametocyte | 43(2.4) | 10(2.8) | 14(3.4) | 05(2.2) | 05(3.1) | 09(1.5) |

## Factors associated with anemia among the study participants

Table 4 presents the multivariate analysis of the factors associated with anemia. Individuals with positive infection by *P. falciparum* were more at risk of anemia (aOR = 1.36, p = 0.027) compared to individuals who were not *P. falciparum* positive. Female individuals were more likely to be anemic (aOR = 2.16, p = 0.000), compared to their male counterparts. Anemia risk significantly decreased as the age of the individual increased; the odds of anemia were 13 times (aOR = 13.01, p = 0.000) higher among under-five children compared to adults aged 20 years old and above. The odds of anemia were around 3 times (aOR = 3.35, p = 0.000) and 2 times higher (aOR = 1.99, p = 0.001) respectively, among children aged 5–9 years old and adolescents aged 10–14 years old, compared to adults 20 years of age and above.

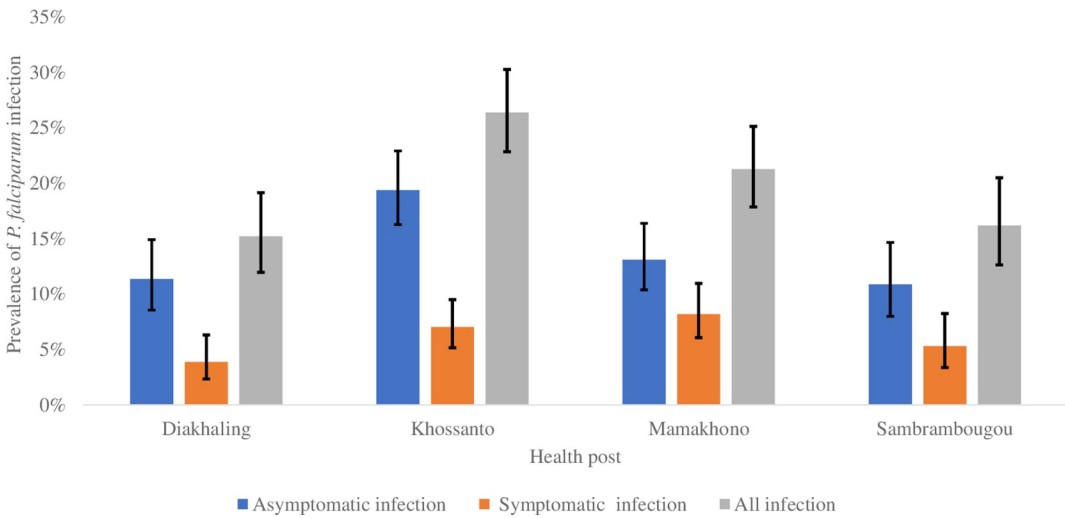

**Fig 3. Prevalence of malaria infection by health post.**

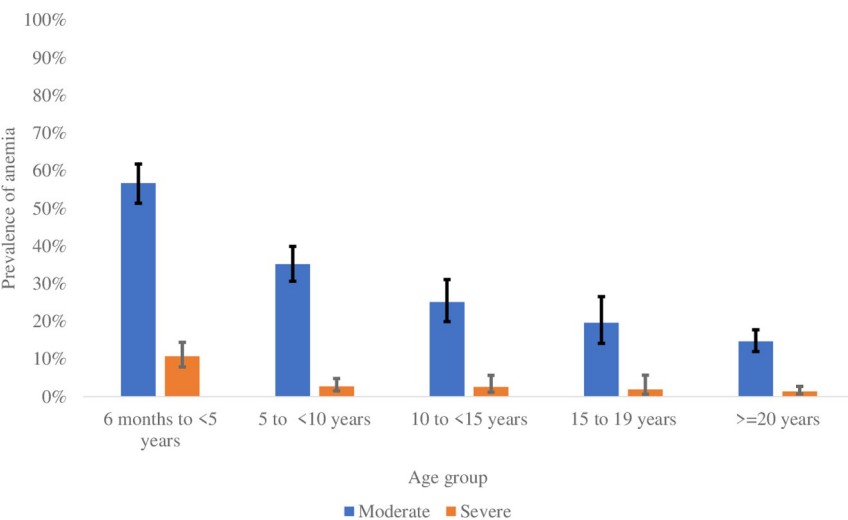

**Fig 4. Prevalence of anemia by age group.**

## Discussion

As countries shift from control to pre-elimination phase, assessing the magnitude of malaria infection in all age groups has become crucial, for a more targeted approach to intervention delivery. To the best of our knowledge, there is limited data on parasite prevalence among adolescents and adults in Senegal. This cross-sectional survey was thus conducted to evaluate the prevalence of malaria infections among a population living in an area with persistent malaria

**Table 3. Estimates and odds ratios (OR) with 95% confidence intervals.**

| Parameters | Estimates | OR | 95% CI | P-value |
|---|---|---|---|---|
| **Intercept** | -2.10 | | | 0.000 |
| **Age (years)** (Ref. reference, age 0.5 to <5) | | | | |
| 5 to <10 | 0.34 | 1.40 | 1.02–1.91 | **0.035** |
| 10 to <15 | 0.25 | 1.28 | 0.88–1.87 | 0.198 |
| 15 to <20 | 0.30 | 1.35 | 0.91–1.99 | 0.135 |
| ≥ 20 | -0.44 | 0.64 | 0.46–0.90 | **0.010** |
| **Gender** (Ref. Female) | | | | |
| Male | 0.18 | 1.19 | 0.88–1.61 | 0.253 |
| **Household head education** (Ref. No education) | | | | |
| Koranic | 0.04 | 1.04 | 0.74–1.48 | 0.807 |
| Primary | 0.23 | 1.26 | 0.92–1.73 | 0.150 |
| Secondary | -0.44 | 0.64 | 0.22–1.92 | 0.430 |
| **Health post** (Ref. Diakhaling) | | | | |
| Khossanto | 0.61 | 1.84 | 1.06–3.20 | **0.030** |
| Mamakhono | 0.22 | 1.25 | 0.78–2.02 | 0.359 |
| Sambrambougou | 0.23 | 1.26 | 0.79–2.02 | 0.337 |
| **Use of bed net the previous night** (Ref. Yes) | | | | |
| No | 0.10 | 1.10 | 0.79–1.54 | 0.555 |
| **Number of household members** (continuous) | 0.007 | 1.01 | 1.00–1.02 | 0.136 |

**Table 4. Logistic regression of factors associated with anemia among the participants in Saraya health district.**

| Parameters | Estimates | OR | 95% CI | P-value |
|---|---|---|---|---|
| **Intercept** | -2.33 | | | **0.000** |
| **Age (years)** (Ref. reference, age > = 20)<br>0.5 to <5 | 2.57 | 13.01 | 9.26–18.28 | **0.000** |
| 5 to <10 | 1.21 | 3.35 | 2.34–4.78 | **0.000** |
| 10 to <15 | 0.69 | 1.99 | 1.34–2.96 | **0.001** |
| 15 to <20 | 0.26 | 1.30 | 0.76–2.23 | 0.343 |
| **Gender** (Ref. Male) | | | | |
| Female | 0.77 | 2.16 | 1.68–2.78 | **0.000** |
| **Malaria parasite** (Ref. Absence) | | | | |
| Presence | 0.31 | 1.36 | 1.04–1.79 | **0.027** |
| **Use of bed net the previous night** (Ref. Yes) | | | | |
| No | 0.22 | 1.25 | 0.92–1.70 | 0.153 |

transmission in Senegal and included adolescents and other understudied groups of the population, to generate evidence to help guide future malaria interventions.

The study revealed a high burden of malaria infection with an overall prevalence of *P. falciparum* infection of 20% among the study participants. Though no statistical difference was found in term of malaria prevalence between children aged under ten and adolescents (aged 10 to 19 years of age), adolescents were found to have a high infection rate, suggesting that adolescents may represent an important source of malaria reservoir and could be potential drivers of malaria infections. Moreover, despite a higher risk of malaria infection, adolescents remained under-protected with the lowest rate of LLIN usage compared to other population groups. Other studies conducted in the same area revealed sub-optimal care-seeking behavior for fever management among adolescents as well as a low uptake of malaria preventive measures [26,27]. These data call for an urgent need to develop specific interventions targeting these new emerging vulnerable groups, that continue to be missed by health care interventions. Most malaria interventions in Senegal, continue to target common at-risk populations such as children under five and pregnant women. Children under 10 are currently being targeted by specific control interventions such as seasonal malaria chemoprevention (SMC), extending SMC to adolescents could be considered by NMCP in Senegal for effective control of malaria infections.

It was noted that most of the *P. falciparum* malaria infections (69.3%) were asymptomatic, with the highest proportion among 15–19 years old adolescents (87.5%) and adults over 20 years. This could be explained by the acquisition of partial immunity, due to recurrent exposure to mosquito bites [28,29]. Asymptomatic infections constitute a threat for malaria control, as such infections can persist for months, specifically among older age groups, and will not be detected and therefore could remain untreated. As a consequence, these individuals serve as parasite reservoirs and can continue to sustain transmission in the community [28,30,31]. In this context, developing new approaches that will target asymptomatic infections alongside existing interventions could be important to accelerate malaria control and elimination. An enhanced community-based surveillance system, or a strategy of screening and provision of treatment could be useful within this epidemiological context. Both of these interventions have been shown to be promising for malaria control and elimination in Senegal [32]. However, the proportion of symptomatic malaria infection was high among small (under five) and school-age (5–14 years old) children, and could be explained by their increased susceptibility to the parasite due to low or lack of immunity [33].

LLIN ownership in the study area remained low in the entire study population with an overall household bed net ownership at 49.5%, which is below the NMCP strategic target of 80% [3]. This may be explained by the fact that the last mass campaign of long-lasting insecticide nets (LLIN) was conducted three years prior to the survey, and the subsequent one was planned at the time of the survey. Studies on the durability of LLIN revealed that the median survival time of LLIN in Senegal was at 2.4 years [34], demonstrating the need for malaria programs to develop alternative strategies that can help maintain high LLIN coverage between mass campaigns. This study also revealed a gap between ownership and use of the LLIN, with only 43% of the participants reporting using the bed net. It appears that there is still a gap between ownership and usage of LLIN as reported by several malaria programs in Sub-Saharan Africa [35]. Operational research could be helpful to understand why there is still a persistent gap between net ownership and net usage despite significant efforts by malaria programs to improve net usage [35].

Malaria exposure seems to be more pronounced in one health post compared to other study areas suggesting a spatial heterogeneity in terms of malaria distribution. Establishing a surveillance system that can provide real-time identification of malaria hotspots is therefore needed. Malaria surveillance in many settings in Senegal relies mainly on aggregate data of malaria cases from routine outpatient visits at health facilities and cross-sectional community household surveys to assess malaria prevalence. Such a system may not be efficient to accurately characterize the disease epidemiology, especially when transmission is so spatially heterogeneous. Further efforts are thus needed to strengthen surveillance systems for a more targeted approach of anti-malaria interventions delivery.

Around one-third (33.5%) of the study participants were anemic, including 89% presenting moderate anemia. The highest anemia rate was observed among children under ten years old, and anemia remained closely associated with malaria infection as described in previous studies [36–40]. However, three-quarters of anemic participants in the present study were *P. falciparum* negative. Anemia etiologies are multifactorial and other factors including soil-transmitted helminthiasis [37,41], and stunting [42,43] could explain the high prevalence of anemia observed among these non-malaria participants but our study did not assess these factors. Prevalence of anemia decreased with an increase in age in our study. Similar results have been shown in other studies [37,39,44,45]. Iron deficiency due to the high demand for iron by the body during the childhood, and/or the loss of iron due to parasitic infections such as malaria and intestinal worms may explain this result [41,46]. Being female was associated with a higher risk of anemia; biological differences between females and males may explain this result [47,48].

### Study limitation

There are a number of limitations to this study. Submicroscopic malaria infections were not assessed. Molecular studies to evaluate their magnitude and their potential contribution to disease transmission might provide additional epidemiological data relevant to the study settings.

The study did not collect data on entomological features and on environmental factors including temperature, humidity, rainfall that may influence malaria distribution. Such data would help to better evaluate the spatial heterogeneity of malaria transmission dynamics.

### Conclusion

The prevalence of *P. falciparum* infection was high among adolescents and children under ten; with adolescents more commonly presenting as asymptomatic. Interventions tailored to adolescents are needed to optimize malaria control. Additional interventions such as extending SMC to adolescents, developing a strategy for screening and treatment of adolescents as well as

an enhanced community-based surveillance for fever case management could be options to consider. Operational research would be useful to assess the feasibility and potential impact of such approaches.

## Supporting information

**S1 Checklist. STROBE statement—checklist of items that should be included in reports of *cross-sectional studies.***
(DOCX)

**S1 File. French version of the study questionnaire.**
(DOCX)

**S2 File. English version of the study questionnaire.**
(DOCX)

**S1 Dataset. Prevalence survey excel dataset.**
(XLSX)

## Acknowledgments

The authors acknowledge Dr Baba Camara, Head of the District Health of Saraya, and his team who facilitated the conduct of the study in the health district, as well as the population of the study area for their participation in the study.

## Author Contributions

**Conceptualization:** Fassiatou Tairou, Roger C. K. Tine.

**Data curation:** Fassiatou Tairou.

**Formal analysis:** Fassiatou Tairou, Ibrahima Gaye, Roger C. K. Tine.

**Investigation:** Fassiatou Tairou, Roger C. K. Tine.

**Methodology:** Fassiatou Tairou, Roger C. K. Tine.

**Supervision:** Fassiatou Tairou, Libasse Sarr, Roger C. K. Tine.

**Validation:** Fassiatou Tairou, Samantha Herrera, Saira Nawaz, Libasse Sarr, Birane Cissé, Babacar Faye, Roger C. K. Tine.

**Writing – original draft:** Fassiatou Tairou.

**Writing – review & editing:** Fassiatou Tairou, Ibrahima Gaye, Samantha Herrera, Saira Nawaz, Libasse Sarr, Birane Cissé, Babacar Faye, Roger C. K. Tine.

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
