## [Decision Letter · Decision Letter 0]

9 Jan 2024

PONE-D-23-41641Malaria prevalence and use of control measures in an area with persistent transmission in SenegalPLOS ONE

Dear Dr. Tairou,

Thank you for submitting your manuscript to PLOS ONE. After careful consideration, we feel that it has merit but does not fully meet PLOS ONE’s publication criteria as it currently stands. Therefore, we invite you to submit a revised version of the manuscript that addresses the points raised during the review process.

We look forward to receiving your revised manuscript.

Kind regards,

Sylla Thiam, M.D, MPH

Academic Editor

PLOS ONE

Journal Requirements:

“The study was supported by the Senegalese National Malaria Control Program, through an agreement with the Department of Parasitology and Mycology of University Cheikh Anta Diop of Dakar (# funding number: NFM 2-SEN-M-PNLP 2018 - 2019 -2020). The funder had no role in the study design, data collection, and analysis, decision to publish, or preparation of the manuscript”

Additional Editor Comments (if provided):

The paper is very relevant and timely. The study is well designed and easily explained. The manuscript is well written in a very simple and accessible language.

Results are well presented with their respective public health implications. Lastly key recommandations are given to strenghten malaria surveillance in all age groupes and support national effort towards malaria elimination.

However, Saraya being a high transmission area where SMC is implemented since almost 10 years, we cannot understand that SMC data was not collected among children under 10 years.

The authors did not give any explanation on the absence of SMC data during data collection and analysis.

The absence of SMC data is a major limit that needs to be adressed by the authors.

Reviewers' comments:

Reviewer's Responses to Questions

**Comments to the Author**

1. Is the manuscript technically sound, and do the data support the conclusions?

Reviewer #1: Partly

Reviewer #2: Yes

2. Has the statistical analysis been performed appropriately and rigorously? 

Reviewer #1: Yes

Reviewer #2: Yes

3. Have the authors made all data underlying the findings in their manuscript fully available?

Reviewer #1: Yes

Reviewer #2: Yes

4. Is the manuscript presented in an intelligible fashion and written in standard English?

Reviewer #1: Yes

Reviewer #2: Yes

5. Review Comments to the Author

Reviewer #1: This study was conducted in a high malaria transmission area of Senegal, the district of Saraya where seasonal malaria chemoprevention (SMC) is implemented for children under 10 years of age. In Saraya district’s, under 10 children receive monthly a curative dose of antimalarial during the malaria transmission season from June to October.

Why don’t the authors include the SMC coverage in their data collection?

Is the low prevalence of asymptomatic and symptomatic malaria in children under 10 not due to SMC?

I think that conducting a malaria prevalence study in an area like Saraya should take into account the coverage rate for a better interpretation of the results.

I recommend that authors include the CMS coverage rate in their data collection.

Corrections are also needed for Plasmodium species. P. falciparum instead of P. Falciparum

Reviewer #2: General Comment: The title of the manuscript is suitable and relevant to the health challenges ravaging the sub-Sahara subregion of the world. The research is well-conceived and executed. The language of communication and the grammar are good. The methodology and data analysis are explicit. The result is well presented in tables and figures. The discussion is well written. References are ok. However, there are a few issues for clarification.

Abstract:

Page 3 Line 50-51 & 56-57. There are discrepancies between the statement on malaria by age group.

Line 50-51. “There was no statistical difference in malaria infection by age group.”

Line 56-57. “Participants aged 5-9 years were more likely to have malaria infection compared to under five children (aOR=1.40, 95% CI:1.02-1.91).”

Introduction:

Page 4. Line 74-78. It was observed that you used different indices for malaria mortality, which did not allow for an easy comparison of the trend in malaria mortality between 2020 and 2022. You are advised to use the same metric for comparison to ensure objectivity.

Method:

Page 6. Line 113. You will need to be explicit on inclusion and exclusion criteria as individuals on antimalaria may be asymptomatic at the time of the recruitment and should have been excluded as they may still harbour the parasite.

Page 7. Line 161-163. Provide a reference for the Senegalese population used to compare the proportion of each age group recruited.

Page 8. Line 183-189. Information on individuals concerned with preparing the smear should be provided. Were they trained?

Page 12. Table 1. The table shows that 43.2% had no occupation, while 25.7% were students. What, then, is the main source of livelihood for most of the populace?

Table1. page 12. Different spellings were used for the “Quranic” school….”Koranic”….”Choranic”. Kindly reconcile the spelling.

Page 14-15. Consistent use of abbreviation for Plasmodium falciparum as P. falciparum. Ensure the same abbreviation after the initial full written form.

Discussion. Ok

Page 22. Line 452-458. The study limitation, which considers the microscopy method used in this community study to be a possibility of missed diagnosis in low-density infection, may need to be revised.

In a study by Berzosa et al. 2018 in Equatorial Guinea, the microscopy used in your research is still recommended by WHO as it is cheap, allows for species differentiation, and is the acclaimed gold standard for diagnosis by WHO.

“Berzosa, P., de Lucio, A., Romay-Barja, M. et al. Comparison of three diagnostic methods (microscopy, RDT, and PCR) for detecting malaria parasites in representative samples from Equatorial Guinea. Malar J 17, 333 (2018). https://doi.org/10.1186/s12936-018-2481-4” three metho

References. Ok

6. PLOS authors have the option to publish the peer review history of their article (what does this mean?). If published, this will include your full peer review and any attached files.

Reviewer #1: No

Reviewer #2: **Yes: **BELLO IBRAHIM SEBUTU

---

## [Author Response · Author response to Decision Letter 0]

24 Mar 2024

To PLOS ONE editor

Subject: Response to Reviewers

Ref: PONE-D-23-41641

Malaria prevalence and use of control measures in an area with persistent transmission in Senegal

Dear Editor;

On behalf of the co-authors, I would like to thank the editor and the reviewers for the comments and submit below the point-by-point responses to the concerns raised during the review process:

https://journals.plos.org/plosone/s/file?id=ba62/PLOSOne_formatting_sample_title_authors_affiliations.pdf"

Authors’ reply: The manuscript has been revised in accordance with PLOS ONE’s style requirements, including those for file naming

“The study was supported by the Senegalese National Malaria Control Program, through an agreement with the Department of Parasitology and Mycology of University Cheikh Anta Diop of Dakar (# funding number: NFM 2-SEN-M-PNLP 2018 - 2019 -2020). The funder had no role in the study design, data collection, and analysis, decision to publish, or preparation of the manuscript”

Authors’ reply: The amended statements (see below) have been included in the cover letter:

The study was supported by the Senegalese National Malaria Control Program, through an agreement with the Department of Parasitology and Mycology of University Cheikh Anta Diop of Dakar (# funding number: NFM 2-SEN-M-PNLP 2018 - 2019 -2020). There was no additional external funding received for this study. The funders had no role in study design, data collection and analysis, decision to publish, or preparation of the manuscript.

Authors’reply: The correct grant number is (NFM 2-SEN-M-PNLP 2018 - 2019 -2020). 

Authors’reply: The manuscript has been revised to reflect separate caption for each figure

Authors’reply: Fig 1 illustrating the study site was created by one of the co-authors (Mr Libasse Sarr) for this article, and was not previously published anywhere. A permission to publish the figure under the Creative Commons Attribution License (CCAL) CC BY 4.0 is requested from him and the Content Permission Form is submitted along with the revised version of the manuscript.

6. Additional Editor Comments (if provided): 

The paper is very relevant and timely. The study is well designed and easily explained. The manuscript is well written in a very simple and accessible language.

Results are well presented with their respective public health implications. Lastly key recommendations are given to strenghten malaria surveillance in all age groupes and support national effort towards malaria elimination. However, Saraya being a high transmission area where SMC is implemented since almost 10 years, we cannot understand that SMC data was not collected among children under 10 years. The authors did not give any explanation on the absence of SMC data during data collection and analysis. The absence of SMC data is a major limit that needs to be adressed by the authors.

Authors’reply: The study collected data on SMC coverage among children under 10 years. These data are presented in the revised version as follow:

According to the report of caregivers/mothers, a proportion of sixty-eight percent (68.8%) of children aged 6 months to 10 years (n=839) received at least one dose of SMC; full coverage of SMC (3 days of treatment for all four rounds) was evaluated at 46.8% (see Table 1, page 13)

Reviewer #1:

1. This study was conducted in a high malaria transmission area of Senegal, the district of Saraya where seasonal malaria chemoprevention (SMC) is implemented for children under 10 years of age. In Saraya district’s, under 10 children receive monthly a curative dose of antimalarial during the malaria transmission season from June to October. 

Why don’t the authors include the SMC coverage in their data collection?

I think that conducting a malaria prevalence study in an area like Saraya should take into account the coverage rate for a better interpretation of the results. I recommend that authors include the CMS coverage rate in their data collection.

Authors’reply : The study collected data on SMC coverage among children under 10 years. These data are presented in the revised version as follow:

According to the report of caregivers/mothers, a proportion of sixty-eight percent (68.8%) of children aged 6 months to 10 years (n=839) received at least one dose of SMC; full coverage of SMC (3 days of treatment for all four rounds) was evaluated at 46.8% (see Table 1, page 13)

2. Is the low prevalence of asymptomatic and symptomatic malaria in children under 10 not due to SMC? 

Authors’reply: Though SMC data was collected in this study, we did not find any evidence of SMC effect on symptomatic or asymptomatic P. falciparum malaria due to lack of power. Indeed, assessing SMC effect on symptomatic and asymptomatic infections, will require subgroup analysis with restrictions on SMC targets who were found with P. falciparum infections either symptomatic or asymptomatic (N=198). This prevalence study was designed to assess malaria prevalence at all age group but was not powered to detect an effect of SMC on symptomatic or asymptomatic malaria.

3. Corrections are also needed for Plasmodium species. P. falciparum instead of P. Falciparum

Authors’reply: Plasmodium species “P. falciparum” has been harmonized throughout the manuscript

Reviewer #2: 

1. Abstract:

Page 3 Line 50-51 & 56-57. There are discrepancies between the statement on malaria by age group. 

Line 50-51. “There was no statistical difference in malaria infection by age group.” 

Line 56-57. “Participants aged 5-9 years were more likely to have malaria infection compared to under five children (aOR=1.40, 95% CI:1.02-1.91).”

Authors’reply: The authors agreed with the reviewer. 

Line 50-51 refer to univariate analysis and this has showed a significant effect of age on malaria prevalence (specifically between children aged 5-10 years old and adults aged �20 years old). To avoid confusion with the multivariate analysis, the manuscript is revised and the sentence in Line 50-51 is removed.

2. Introduction:

 Page 4. Line 74-78. It was observed that you used different indices for malaria mortality, which did not allow for an easy comparison of the trend in malaria mortality between 2020 and 2022. You are advised to use the same metric for comparison to ensure objectivity.

Authors’reply: This section has been revised in the manuscript. “Mortality rate” is used to compare malaria mortality between 2020-2022 (Please, see Line 78-80)

3. Method:

Page 6. Line 113. You will need to be explicit on inclusion and exclusion criteria as individuals on antimalaria may be asymptomatic at the time of the recruitment and should have been excluded as they may still harbour the parasite.

Authors’reply: The manuscript has been revised to take into account the reviewer’s comment; inclusion and exclusion criteria considered during the recruitment are added to the revised manuscript (Please, see Line 126-130).

Page 7. Line 161-163. Provide a reference for the Senegalese population used to compare the proportion of each age group recruited.

Authors’reply: The reference for the Senegalese population used for the proportion of each age group is provided in the revised manuscript (Please, see Line 172).

Page 8. Line 183-189. Information on individuals concerned with preparing the smear should be provided. Were they trained?

Authors’reply: Individuals involved in the preparation of the smear were trained on the sample collection and the preparation of the smears. The manuscript is revised to reflect this information Please, see Line 201-204).

Page 12. Table 1. The table shows that 43.2% had no occupation, while 25.7% were students. What, then, is the main source of livelihood for most of the populace?

Authors’reply: In the study area, the main source of livelihood for most of the population is farming and gold mining . This figure is well represented among adults in the study population, with around 60% (farming 21.2% and gold mining 38.4%) (Please, see Table 1).

Table1. page 12. Different spellings were used for the “Quranic” school….”Koranic”….”Choranic”. Kindly reconcile the spelling.

Authors’reply: The manuscript is revised and “Koranic school” is used throughout the document.

Page 14-15. Consistent use of abbreviation for Plasmodium falciparum as P. falciparum. Ensure the same abbreviation after the initial full written form.

Authors’reply: The manuscript is revised in accordance. P. falciparum has been used throughout the manuscript as abbreviation for Plasmodium falciparum.

4. Discussion. 

Page 22. Line 452-458. The study limitation, which considers the microscopy method used in this community study to be a possibility of missed diagnosis in low-density infection, may need to be revised.

 In a study by Berzosa et al. 2018 in Equatorial Guinea, the microscopy used in your research is still recommended by WHO as it is cheap, allows for species differentiation, and is the acclaimed gold standard for diagnosis by WHO. 

“Berzosa, P., de Lucio, A., Romay-Barja, M. et al. Comparison of three diagnostic methods (microscopy, RDT, and PCR) for detecting malaria parasites in representative samples from Equatorial Guinea. Malar J 17, 333 (2018). https://doi.org/10.1186/s12936-018-2481-4” three metho

Authors’reply: The authors would like to thank the reviewer for this comment. This limitation has been removed from the revised manuscript (Please, see Line 512-513).

---

## [Editor Report · Decision Letter 1]

1 May 2024

Malaria prevalence and use of control measures in an area with persistent transmission in Senegal

PONE-D-23-41641R1

Dear Dr. Tairou

We’re pleased to inform you that your manuscript has been judged scientifically suitable for publication and will be formally accepted for publication once it meets all outstanding technical requirements.

Kind regards,

Sylla Thiam, M.D, MPH

Academic Editor

PLOS ONE